# Influence of running shoes on muscle activity

**Fabian Hoitz**[1,2]*, **Jordyn Vienneau**[2], **Benno M. Nigg**[2]

**1** Biomedical Engineering, Schulich School of Engineering, University of Calgary, Calgary, Alberta, Canada,
**2** Human Performance Laboratory, Faculty of Kinesiology, University of Calgary, Calgary, Alberta, Canada

* fabian.hoitz@ucalgary.ca

## Abstract

Studies on the paradigm of the preferred movement path are scarce, and as a result, many aspects of the paradigm remain elusive. It remains unknown, for instance, how muscle activity adapts when differences in joint kinematics, due to altered running conditions, are of low / high magnitudes. Therefore, the purpose of this work was to investigate changes in muscle activity of the lower extremities in runners with minimal ($\leq 3°$) or substantial ($> 3°$) mean absolute differences in the ankle and knee joint angle trajectories when subjected to different running footwear. Mean absolute differences in the integral of the muscle activity were quantified for the tibialis anterior (TA), peroneus longus (PL), gastrocnemius medialis (GM), soleus (SO), vastus lateralis (VL), and biceps femoris (BF) muscles during over ground running. In runners with minimal changes in 3D joint angle trajectories ($\leq 3°$), muscle activity was found to change drastically when comparing barefoot to shod running (TA: 35%; PL: 11%; GM: 17%; SO: 10%; VL: 27%; BF: 16%), and minimally when comparing shod to shod running (TA: 10%; PL: 9%; GM: 13%; SO: 8%; VL: 8%; BF: 12%). For runners who showed substantial changes in joint angle trajectories ($> 3°$), muscle activity changed drastically in barefoot to shod comparisons (TA: 39%; PL: 14%; GM: 16%; SO: 16%; VL: 25%; BF: 24%). It was concluded that a movement path can be maintained with small adaptations in muscle activation when running conditions are similar, while large adaptations in muscle activation are needed when running conditions are substantially different.

## Introduction

In the last four decades, scientific discussions on running biomechanics and running injuries have been dominated by two paradigms: the "impact force" paradigm and the "pronation" paradigm [1]. In short, these paradigms suggest that higher magnitudes of impact forces and / or pronation that may occur during running are harmful to the human body and may lead to the development of running injuries. Consequently, advancements in running shoes, shoe inserts, and orthotics have aimed to reduce impact forces [2], and / or to re-align ankle kinematics [3]. Despite the vast financial investment in the development of these products, however, running injury rates have remained relatively unchanged [4–6]. This lack of epidemiological evidence led recent publications to question the validity of the these paradigms, arguing that they were derived from an inappropriate functional understanding of running biomechanics [7]. As a result, new paradigms have been proposed, aiming to redirect future studies to the functional

**Funding:** This study was funded by Mizuno Corporation (Osaka, Japan) and Biomechanigg Sport and Health Research (Calgary, Canada).

Mizuno Corporation (Osaka, Japan) also provided the shoes that were used in the testing. However, the results presented in this article do not in any way represent a bias toward Mizuno products over other brands. The results of the study are also presented clearly, honestly, and without fabrication, falsification, or inappropriate data manipulation. The funders had no role in study design, data collection and analysis, decision to publish, or preparation of the manuscript.

**Competing interests:** This work was funded by Mizuno Corporation (Osaka, Japan) and Biomechanigg Sport and Health Research (BSHR; Calgary, Canada). Mizuno Corporation (Osaka, Japan) also provided the shoes that were used in the testing. However, the results presented in this article do not in any way represent a bias toward Mizuno products over other brands. The principal investigator, Dr. Benno Nigg, is also the Chief Science Officer of the sponsoring company BSHR. BSHR covered material costs for this research and was simply interested in the outcome of the study, regardless of the findings. The company BSHR did not benefit from the results of the findings. BSHR had no influence on the outcome of this study. The results of the study were presented clearly, honestly, and without fabrication, falsification, or inappropriate data manipulation. The funders had no role in study design, data collection and analysis, decision to publish, or preparation of the manuscript. This does not alter our adherence to PLOS ONE policies on sharing data and materials.

aspects of running, by focusing on the effect of internal forces, their influence on running biomechanics, and how they can be impacted by different running shoes [1, 8, 9]. It is important to note that these novel paradigms do not suggest that the interpretation and analysis of traditional variables (e.g., ground reaction force, joint kinematics) is frivolous. Instead, these novel paradigms aim to provide new perspectives on running biomechanics that are based on a functional understanding of running.

One of these newly proposed paradigms–the preferred movement path paradigm–suggests that runners are likely to maintain a consistent movement path (i.e., movement trajectories) when changing between reasonably similar shoes (e.g., cushioned shoe vs. motion control shoe). It was speculated that the locomotor system aims to maintain this preferred movement path as it may be associated with reduced energy demands, lower joint and tissue loading, and / or lower risk of injury [10]. Potential implications have been investigated by a recent study [11] that showed that the loss in cartilage volume after a prolonged run could be reduced in runners who wore footwear that facilitated a runner's natural joint motion. Consequently, footwear constructions that do not support a preferred movement path may be harmful to the locomotor system and may potentially cause an increased energy / muscle activity demand, and / or an increased risk of injury. The preferred movement path of a given runner, however, is not expected to be constant. Rather, it may depend on factors such as fatigue, training status, the presence of injury, and / or substantial changes in footwear constructions. For instance: a preferred movement path may be different in a running shoe compared to a worker's boot. It was reported, for example, that more than 80% of runners exhibited changes of less than 3° in ankle and knee joint kinematics when running in two similar shoe conditions [10]. Conversely, for a more dramatically different comparison (running barefoot vs. shod), most participants (91%) changed their segment trajectories by more than 3°. It appears, therefore, that small changes in footwear constructions do allow runners to maintain a consistent movement path, while larger modifications may force adaptations in gait patterns.

Many aspects of the preferred movement path remain elusive. It is unclear, for instance, how the locomotor system is able to maintain a consistent movement path despite changing footwear constructions. Furthermore, the role of footwear constructions with respect to their beneficial and / or detrimental effects on a runner's preferred movement path remains unknown.

It has been proposed that muscle activation patterns play an important role in the underlying principles that govern a runner's preferred movement path [10]. One can speculate that adaptations in muscle activity would allow the locomotor system to maintain a movement path that is preferred when boundary conditions (e.g., footwear constructions, occurrence of injuries, etc.) change. Consequently, footwear constructions that reduce muscular activity without forcing a runner to change their preferred movement path may be beneficial (i.e., reduce injury risks and / or energy demands). However, when the locomotor system is forced to adopt a novel preferred movement path, such as when changing from barefoot to shod running (where kinematic changes are substantial), one would expect muscle activity to change drastically, in order to accommodate this new situation. Previous studies already highlighted some changes in muscle activation when comparing barefoot to shod running [12, 13]. During barefoot running, for example, the activity of the plantarflexors (gastrocnemius medialis / lateralis, and soleus) was shown to increase before heel strike [14] and the tibialis anterior has been shown to increase during the stance phase [15].

It appears evident, therefore, that muscle activation strategies are altered when kinematic differences are substantial. These outcomes, however, have yet to be investigated through the lens of the preferred movement path paradigm. It is currently unknown how muscle activation changes when a movement path is maintained (i.e., small kinematic differences) as opposed to

when a novel movement path is adopted (i.e., large kinematic differences). As a result, the purpose of this work was to investigate changes in lower extremity muscle activation in runners with minimal or substantial ($\leq 3°$ or $> 3°$) mean absolute differences in the ankle and knee joint angle trajectories when subjected to different running footwear. Specifically, mean absolute changes in the integral of muscle activation for the tibialis anterior (TA), peroneus longus (PL), gastrocnemius medialis (GM), soleus (SO), vastus lateralis (VL), and biceps femoris (BF) were quantified in six footwear comparisons.

## Methods

### Participants

Thirty-three heel-toe runners ([mean ± SD]: 17 men: age 31.6 ± 9.9 yrs, mass 77.3 ± 9.0 kg; and 16 women: age 28 ± 9.9 yrs, mass 60 ± 7.6 kg) took part in this study. The focus was placed on heel-to-toe running as it represents the dominant foot strike pattern amongst runners [16]. All participants were healthy (injury free for at least 6 months) and physically active recreational runners (at least 2 runs per week). The average running distance by each participant for any given run was not collected but was estimated to be between 5 and 10 km, according to conversations with the participants. All runners gave written informed consent prior to participation. This study was reviewed and approved by the University of Calgary's Conjoint Health Research Ethics Board under the number REB13-0275.

### Protocol

Testing took place on a single day in an indoor laboratory at the Human Performance Laboratory of the University of Calgary. Participants performed ten running trials (approx. 10 steps per trial) at 3.3 m/s (± 15%) in three shoe conditions that varied in their material properties (Fig 1, Table 1) and barefoot along a 30 m runway. These footwear models were selected to represent a wide range of available footwear solutions, namely a minimalist (Be), a conventionally cushioned (Rider), and a racing flat (Universe) shoe. An important difference between the designs of the Universe and Be was that the Universe had a flat, thin outer sole with a middle groove on the outer sole heel, whereas the Be design incorporated a round outer sole and a gap space under the toe area. Each shoe model was available in multiple sizes, and two pairs were available in each size and condition. Therefore, each test shoe was either new, or had been worn at-most by two previous participants. The four running conditions were tested in a randomized order. Special care was taken to ensure that participants remained in their habitual rearfoot running style in all conditions by monitoring the runner directly and by confirming the presence of an impact peak and heel strike in the force and motion data.

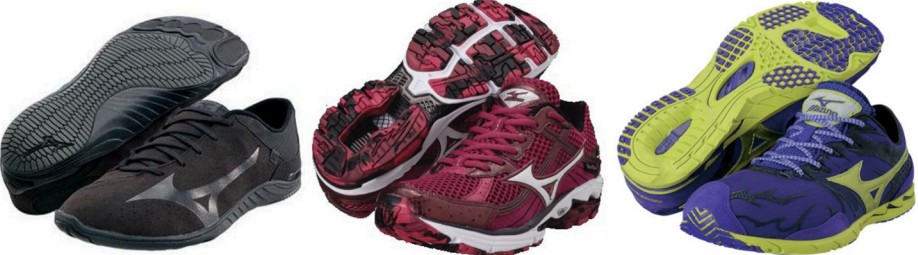

**Fig 1. Evaluated running shoes.** The running shoe models evaluated in this study were the Mizuno Be (left), the Mizuno Wave Rider (centre), and the Mizuno Wave Universe (right).

**Table 1. Physical characteristics for each of the three shoe conditions for men's US size 9.**

|  | Be | Rider | Universe |
|---|---|---|---|
| Midsole (EVA) hardness | Shore C, 70 ± 4C | Shore C, 55 ± 4C | Shore C, 56 ± 4C |
| Outer sole (rubber) hardness | Shore A, 60 ± 3C | Shore A, 70 ± 3A | Shore A, 70 ± 3A |
| Heel cushioning (G Score) | 22.7G | 11.2G | 19.4G |
| Mass (g) | 193 | 270 | 112 |
| Heel-drop (mm) | < 3 | 14.1 | 3 |
| Heel outer sole groove width (cm) | N/A | 2.7 | 2.5 |
| Heel outer sole groove distance from heel edge (cm) | N/A | 3.0 | 2.3 |

## Instrumentation

Three-dimensional (3D) marker trajectories of 16 retroreflective markers were collected using an eight-camera motion analysis system (Motion Analysis Corporation, Santa Rosa, CA, USA) operating at a sampling rate of 240 Hz. Following a previously reported setup [10], markers were placed on the right forefoot (3), hindfoot (3), shank (3), thigh (3), and on the right and left anterior and posterior superior iliac spine (4). An additional seven markers were placed on the first and fifth metatarsal, the medial and lateral malleoli and femoral epicondyles, and on the greater trochanter of the right leg to collect data for a neutral standing trial. The data of the standing trial were used to define segment coordinate systems based on the anatomical landmarks and the additional seven markers were removed for the subsequent running trials. A single force plate (Kistler, 9281CA) was synchronised with the motion analysis system and collected ground reaction force data at 2400 Hz. Additionally, timing lights were placed 1.9 m apart along the runway to monitor running speed.

In addition to the kinematic and kinetic recordings, surface electromyography (EMG) data were collected at a sampling frequency of 2400 Hz from the muscle bellies of the tibialis anterior (TA), peroneus longus (PL), gastrocnemius medialis (GM), soleus (SO), vastus lateralis (VL), and biceps femoris (BF) of the same leg, using bipolar Ag-AgCI surface electrodes (Norotrode Myotronics-Noromed Inc., Kent, WA, USA) with a diameter of 10 mm and an inter electrode spacing of 22 mm. Prior to applying the electrodes, the skin surface was shaved, slightly abraded using sand paper and cleaned with an isopropyl wipe. All electrodes were placed parallel to the direction of the underlying muscle fibres based on the SENIAM guidelines [17].

Finally, a one-dimensional (1D) accelerometer (ADXL 78, Analog Devices USA) with a measuring range of ± 50 g, and sampling at 2400 Hz was placed on the right heel and synchronized with the EMG recordings in order to detect heel strike (HS) events. A HS was defined as the first peak in acceleration due to ground impact.

## Data analysis

Prior to any analysis, all data (kinematic and EMG) were visually inspected to ensure data integrity and remove trials that displayed artifacts. Specifically, running trials that did not show a clear rearfoot strike pattern (determined via visual inspection of kinematic data) and EMG signals with movement artifacts (determined by high intensities in the lower frequencies of the power spectrum) were removed from further analyses. As a result, the number of trials included in the analysis varied across participants. However, a minimum of five trials per running condition was ensured. Subsequently, kinematic and EMG data were analysed separately. Resulting kinematic marker trajectories and EMG intensity signals were then compared between all running conditions (Barefoot vs. Rider / Be / Universe, Rider vs. Be, Rider vs. Universe, Be vs. Universe).

Analysis of kinematic data was performed as described in [10]. Specifically, Cortex (Motion Analysis) and Visual 3D (C-Motion Inc., Germantown, MD) were used to process kinematic and kinetic data. Marker trajectories were filtered using a fourth-order low-pass Butterworth filter with a cut-off frequency of 10 Hz. Subsequently, 3D joint angles of the ankle and knee were calculated as the relative rotation between the thigh and shank segments and the shank and hindfoot segments, respectively, using a X-Y-Z Cardan rotation sequence. All joint angles were expressed relative to the standing posture, and temporally normalised to stance phase. Stance phase was defined as the period between touch down and toe-off, which were identified using a 10 N threshold in the vertical ground reaction force. Finally, using custom written Matlab scripts, absolute differences in kinematic movement trajectories were calculated and averaged for the ankle / knee joint over the time-normalised stance phase (0–100%) for each participant and comparison (Figs 2 and 3). For this study, runners were grouped into those who displayed mean absolute differences in movement trajectories below or equal to 3˚, and runners with mean absolute differences in movement trajectories above 3˚, representing a conservative threshold for clinical relevance as suggested in [10].

EMG data were processed using a custom written Matlab script to analyse the same step as the kinematic data, thus enabling comparisons between the two data sets. A window of 300 ms (i.e., 150 ms before to 150 ms after HS) was analysed for all participants.

For each step and muscle, the raw EMG signal was exposed to a wavelet transform with 13 non-linearly scaled wavelets (centre frequencies: 6.9, 19.3, 37.7, 62.1, 92.4, 128.5, 170.4, 218.1, 271.5, 330.6, 395.4, 465.9, 542.1 Hz) to represent the signal in time-frequency space [18, 19]. Then,

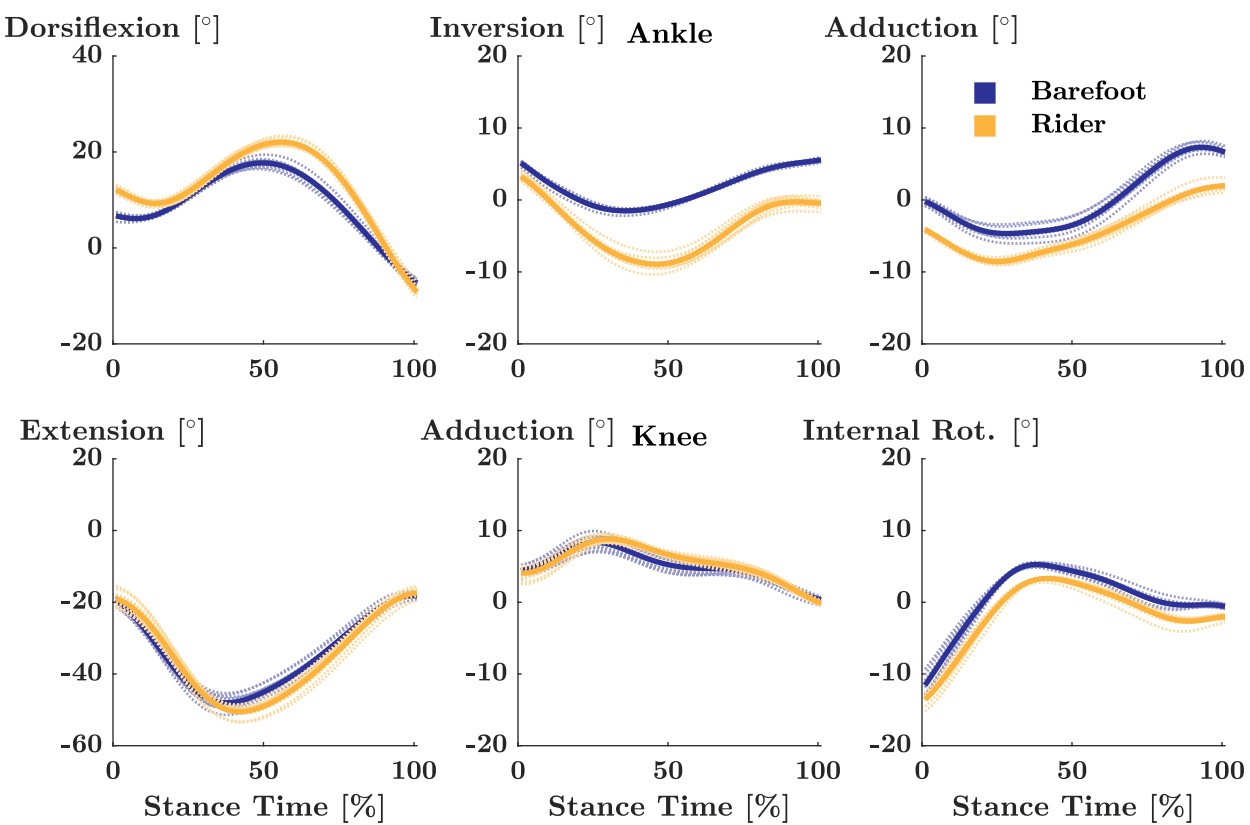

**Fig 2. Exemplary ankle and knee joint kinematics in Barefoot and Rider.** Time normalized mean (solid) and individual (dotted) joint kinematics for the ankle (top) and knee (bottom) of a representative participant in barefoot (blue) and rider (yellow).

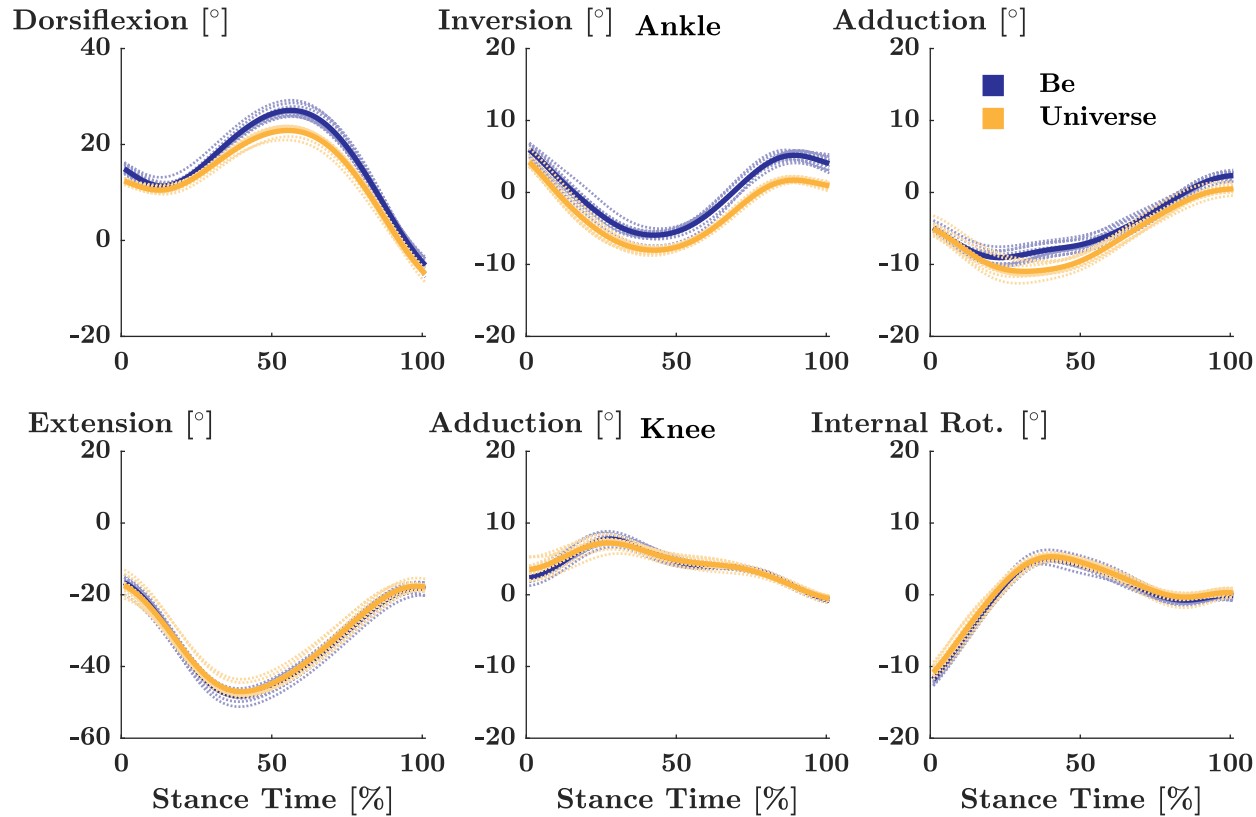

**Fig 3. Exemplary ankle and knee joint kinematics in Be and Universe.** Time normalized mean (solid) and individual (dotted) joint kinematics for the ankle (top) and knee (bottom) of a representative participant in be (blue) and universe (yellow).

each EMG signal was normalised to the sum of the wavelets above 100 Hz (wavelets 6 to 13) of the mean of the barefoot condition. This step reduced the effect of potential movement artifacts, which are associated with lower frequency components (< 100 Hz). Subsequently, the square root of the normalised time-frequency space was summed across all frequencies to obtain the respective EMG intensity signal, of which the area under the curve (AUC) was calculated. For each participant, the mean absolute differences in the AUC were then calculated across all six comparisons (Barefoot vs Rider / Be / Universe, Rider vs Be, Rider vs Universe, Be vs Universe) and each muscle (TA, PL, GM, SO, VL, BF). The outcome was then expressed as a percentage with respect to the first of the two running conditions in each comparison (i.e., Barefoot, Rider, or Be).

## Statistics

Wilcoxon signed-rank tests with a Bonferroni-Holm correction were used to analyse changes in the AUC in runners who showed minimal / substantial ($\leq 3°$ / $> 3°$) differences in 3D joint angle trajectories stratified by six possible footwear comparisons. An obtained p-value smaller than the corrected alpha level indicated significant changes in muscle activation in a given comparison of running conditions.

## Results

The average proportion of participants with mean absolute differences in joint kinematics of $\leq 3°$ and $> 3°$ across barefoot to shod comparisons were 57% and 43%, respectively

**Table 2. Number of participants stratified by comparisons.**

|  | ≤ 3˚ | > 3˚ |
|---|---|---|
| **Barefoot vs Shod** |  |  |
| Barefoot vs Rider | 16 | 17 |
| Barefoot vs Be | 21 | 12 |
| Barefoot vs Universe | 19 | 14 |
| **Shod vs Shod** |  |  |
| Rider vs Be | 33 | 00 |
| Rider vs Universe | 33 | 00 |
| Be vs Universe | 33 | 00 |

Number of participants (N = 33) with kinematic differences of ≤ 3˚ and > 3˚, stratified by running condition comparisons.

(Table 2: barefoot vs shod). In shod to shod comparisons, on average 100% of runners had mean absolute differences in joint kinematics of ≤ to 3˚ (Table 2: shod vs shod), while no runner changed their average joint kinematics by more than 3˚.

In runners with kinematic differences of ≤ 3˚ across running comparisons mean absolute differences in the AUC across all muscles were 19% for barefoot to shod comparisons and 10% for shod to shod comparisons (Fig 4). Specifically, for the barefoot to shod comparisons, the mean absolute differences in the TA, PL, GM, SO, VL, and BF were 35%, 11%, 17%, 10%, 27%, and 16%, respectively. The activity of the TA differed significantly when comparing Barefoot to Be and when comparing Barefoot to Universe ($p < 0.001$ for both). The activity of the VL was significantly different when comparing Barefoot to Rider ($p = 0.001$). When comparing between shoe conditions, differences in EMG were substantially smaller. On average, absolute differences in the TA, PL, GM, SO, VL, and BF were 10%, 9%, 13%, 8%, 8%, and 12%, respectively.

Kinematic differences of more than 3˚ were only observed in Barefoot to Shod comparisons (Table 2). In these comparisons, the mean absolute differences in AUC across all muscles was 12% (Fig 5). The mean differences stratified by muscles were 39%, 14%, 16%, 16%, 25%, and 24% for the TA, PL, GM, SO, VL, and BF respectively. In all three Barefoot to Shod comparisons (Barefoot vs Rider / Be / Universe) the differences in the activity of the TA and VL were significant (TA: $p < 0.001$, $p = 0.002$, $p = 0.001$; VL: $p = 0.001$, $p = 0.002$, $p = 0.002$). In the Barefoot to Universe comparison only, the changes observed in the GM were also significant ($p = 0.002$).

## Discussion

The paradigm of the preferred movement path has been proposed as a replacement for traditional paradigms (i.e., impact force, pronation). It aims to provide a novel perspective on running biomechanics that is based on a functional understanding of running [1]. Many aspects of the preferred movement path paradigm, however, remain disputed and unclear [20, 21]. As a result, its concept will be outlined first in order to discuss the findings of this work within the scope of the novel paradigm.

The paradigm of the preferred movement path suggests that individuals who perform a given task (e.g., running, jumping, etc.), subconsciously adopt a movement pattern (i.e., kinematic movement trajectories) that is preferred under the current set of constraints / boundary conditions (i.e., training status, footwear, etc.). This preferred movement pattern (or movement path) is thought to be the optimal solution (or at least very close to it) for the given task

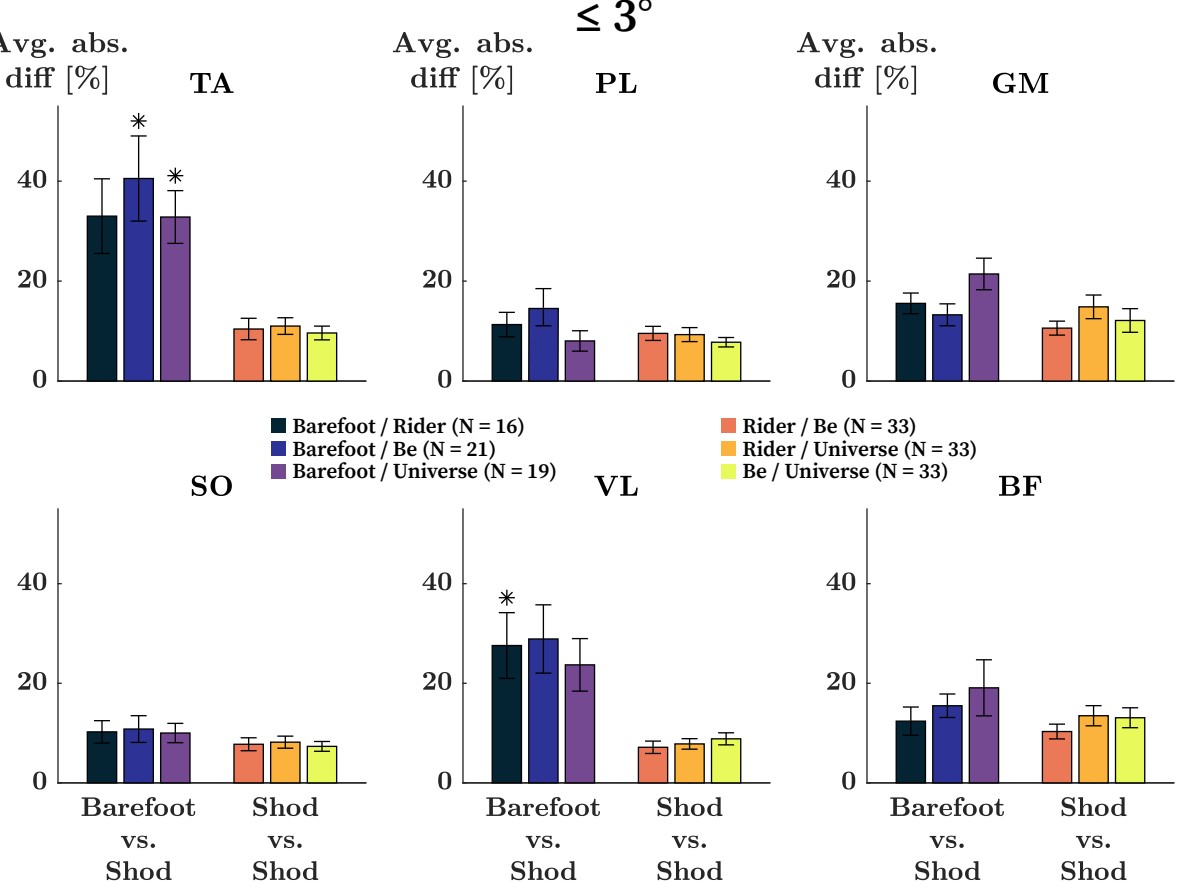

**Fig 4. Changes in EMG activity in runners with kinematic differences ≤ 3˚.** Mean absolute differences in the integral of EMG signals of the tibialis anterior (TA), peroneus longus (PL), gastrocnemius medialis (GM), soleus (SO), vastus lateralis (VL), and biceps femoris (BF) in runners with differences in joint kinematics ≤ 3˚. * Significantly different integrals in the given comparison (p-value ≤ 0.002).

[8]. In other words, the locomotor system fine-tunes its internal parameters (i.e., muscle activation) to perform the task at hand in an optimal way. It is important to note here, that optimal does not exclusively mean most economical (i.e., reduced energy consumption). Instead, the locomotor system aims to optimize for multiple factors. Possible optimization criteria might including an increased feeling of comfort, a reduction in perceived pain, and / or a reduced risk of injury in addition to a reduction in energy consumption. As a result, the solution to this optimization problem is the preferred movement path.

While it is currently unknown how to determine a preferred movement path before a task execution, it has been speculated that observing changes in movement patterns (i.e., joint angle trajectories) may allow researchers to determine whether the preferred movement paths were similar in different interventions [10]. Following this notion, small kinematic deviations may be interpreted as the same movement path across interventions, while larger kinematic deviations may be interpreted as different preferred movement paths. For the present work, a threshold of 3˚ was applied to the mean absolute difference in kinematic movement trajectories across running comparisons. This threshold was selected as it represents a conservative threshold for clinical relevance and was suggested in previous work [10]. Therefore, runners who changed their movement pattern by less than or exactly 3˚ were considered to have had the same movement path across interventions, while runners who changed their movement

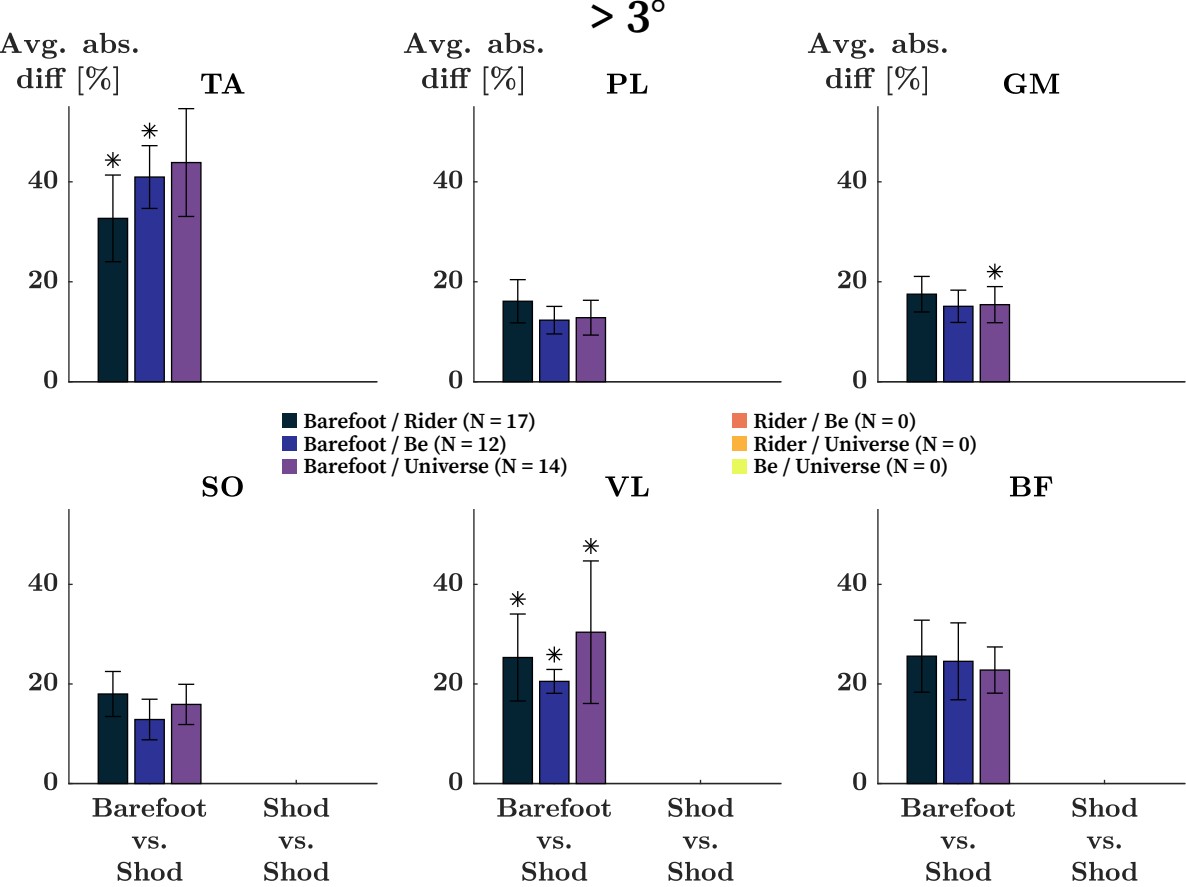

**Fig 5. Changes in EMG activity in runners with kinematic differences > 3˚.** Mean absolute differences in the integral of EMG signals of the tibialis anterior (TA), peroneus longus (PL), gastrocnemius medialis (GM), soleus (SO), vastus lateralis (VL), and biceps femoris (BF) in runners who showed differences in joint kinematics > 3˚. * Significantly different integrals in the given comparison (p-value ≤ 0.002).

pattern by more than 3˚ might have selected a novel (more preferred) movement path for the new intervention.

For both groups (≤ 3˚ and > 3˚), the paradigm of the preferred movement path holds specific implications with respect to the tuning of internal parameters (i.e., muscle activation) in a situation where constraints (i.e., footwear) were altered, but the task (i.e., running at 3.3 m/s) remained the same. For instance, when constraints are altered marginally (i.e., shod to shod comparisons), one would expect small adaptations in internal parameters in runners who maintained the same movement path (≤ 3˚). When constraints are altered drastically (i.e., barefoot to shod comparisons), however, one would expect large adaptations in internal parameters, as that is the only way a runner could maintain the same movement path in the novel situation. For runners who adopt a new preferred movement path (> 3˚), one would expect largely altered internal parameters, when constrains remained similar, but also when constrains are altered drastically.

In the present work, participants were asked to run over-ground at 3.3 m/s in four different running conditions (Barefoot / Rider / Be / Universe). With regard to the preferred movement path paradigm, this describes the same task with altered constraints. Comparisons between footwear conditions (i.e., Rider vs Be, etc.) are considered small changes in constraints, while comparisons between barefoot and shod describe large changes in constraints. As such, the

outcomes of this study support the above outline speculations: small adaptations in EMG in runner who maintained a movement path when switching between running shoes (Fig 4; Shod to Shod), large adaptations in EMG in runners who maintained a movement path but switched between barefoot and shod (Fig 4; Barefoot to Shod), and large adaptations in EMG in runners who adopted a novel preferred movement path (Fig 5).

While these outcomes strengthen the background of the preferred movement path paradigm, they did not provide any explanation as to why some runners maintained a preferred movement path and others did not, despite an identical task. From a functional perspective one could speculate that based on running experience (or expertise) some runners would be more / less willing to adopt a novel movement path. Specifically, more experienced runners would be less likely to change a preferred movement pattern (even under vastly different constraints) because their current movement pattern is as close to the optimal solution as possible. Conversely, in less experienced runners there might be a slightly more optimal solution for the given task and by adopting a novel preferred movement path they perform the movement in a more optimal fashion. The experience level of the runners who participated in this study was, unfortunately, not quantified. Stratifying the response of runners based on experience level should therefore be considered in future investigations.

From a methodological perspective, the selection of a 3° threshold may present some limitations as a threshold indicating a transition to a novel preferred movement path may be runner-specific rather than global. Previous work, for instance, has shown that joint movements, which result in the least amount of resistance are highly variable amongst individuals and specific to a given specimen [22, 23]. Additionally, this study combined deviations in ankle and knee joint kinematics in all three planes within a single measurement, while it has been shown that certain joint components comply better with the concept of the preferred movement path than others [10]. Further, it can be argued that the mean absolute difference in joint trajectories is not an adequate measure of change. While the current study followed a previous example [10], it would be interesting to explore other methodologies to stratify kinematic responses. A comparison across multiple methodologies, for example, may provide strengthening evidence to the paradigm. Future studies are therefore advised to revaluate how to determine deviations from a preferred movement path.

Interpretations from this study should be done with considerations to the following limitations. First, participants were not given an adaptation period after switching between the running conditions. An extended adaptation period may have resulted in smaller deviations in joint angle trajectories, ultimately, reducing the number of participants who selected a novel preferred movement path in the new running condition. Considering, however, that all participants showed minimal ($\leq 3°$) differences in joint kinematics across all Shod to Shod comparisons, this would strengthen the perspective of the paradigm. With respect to Barefoot to Shod comparisons, a reduction in participants who changed their preferred movement path would indicate that running Barefoot is not as different from running Shod as initially speculated. To scrutinize this speculation, future studies might explore the effect of prolonged adaptation periods on changes in joint kinematics and investigate more drastic footwear constructions (i.e., worker's boot, barefoot, running shoe, etc.). Second, the outcomes of this study have been discussed under the light of the preferred movement path paradigm. While the paradigm does explain the outcomes of this study, the paradigm itself is not widely accepted. The present work and the majority of research regarding the paradigm was performed by the research team of Dr. Benno Nigg. This fact highlights a potential research bias with respect to the paradigm and it is possible the findings of the present work could also be interpreted differently. Finally, it was speculated that the locomotor system aims to optimize for multiple factors (i.e., energy consumption, comfort, etc.). The present work, however, did not assess any of these

possible optimization factors and does not provide any evidence for this speculation. Future research is, therefore, advised to incorporate an assessment of possible optimization factors when investigating the paradigm of the preferred movement path.

## Conclusion

A movement path can be maintained with small adaptations in muscle activation when running conditions are similar, while large adaptations in muscle activation are needed when running conditions are drastically different. When a movement path is not maintained, adaptations in muscle activation are drastic.

## Supporting information

**S1 Data.**
(MAT)

## Acknowledgments

We are grateful to Amiée C. Smith for her initial work on this topic and for collecting the data presented in this work.

## Author Contributions

**Formal analysis:** Fabian Hoitz.

**Funding acquisition:** Benno M. Nigg.

**Methodology:** Fabian Hoitz.

**Project administration:** Jordyn Vienneau.

**Resources:** Jordyn Vienneau, Benno M. Nigg.

**Supervision:** Jordyn Vienneau, Benno M. Nigg.

**Visualization:** Fabian Hoitz.

**Writing – original draft:** Fabian Hoitz.

**Writing – review & editing:** Fabian Hoitz, Jordyn Vienneau, Benno M. Nigg.

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
