## [Decision Letter · Decision Letter 0]

16 Apr 2020

PONE-D-20-04762

Influence of running shoes on muscle activity

PLOS ONE

Dear Mr. Hoitz,

Thank you for submitting your manuscript to PLOS ONE. After careful consideration, we feel that it has merit but does not fully meet PLOS ONE’s publication criteria as it currently stands. Therefore, we invite you to submit a revised version of the manuscript that addresses the points raised during the review process.

We would appreciate receiving your revised manuscript by May 31 2020 11:59PM. To enhance the reproducibility of your results, we recommend that if applicable you deposit your laboratory protocols in protocols.io, where a protocol can be assigned its own identifier (DOI) such that it can be cited independently in the future. For instructions see: http://journals.plos.org/plosone/s/submission-guidelines#loc-laboratory-protocols

We look forward to receiving your revised manuscript.

Kind regards,

Jean L. McCrory, PhD

Academic Editor

PLOS ONE

2. Please ensure that you refer to Figure 3 in your text as, if accepted, production will need this reference to link the reader to the figure.

Reviewers' comments:

Reviewer's Responses to Questions

**Comments to the Author**

1. Is the manuscript technically sound, and do the data support the conclusions?

Reviewer #1: Partly

Reviewer #2: Yes

Reviewer #3: Yes

2. Has the statistical analysis been performed appropriately and rigorously? 

Reviewer #1: No

Reviewer #2: Yes

Reviewer #3: No

3. Have the authors made all data underlying the findings in their manuscript fully available?

Reviewer #1: No

Reviewer #2: Yes

Reviewer #3: No

4. Is the manuscript presented in an intelligible fashion and written in standard English?

Reviewer #1: Yes

Reviewer #2: Yes

Reviewer #3: Yes

5. Review Comments to the Author

Reviewer #1: Summary: the authors sought to differentiate those who would alter their kinematics when exposed to different running shod vs. barefoot, and then explore the impact of different shoes on muscle activation in those with altered (>3deg) vs. those minimally altered (<3deg). Those with larger alterations in kinematics experienced larger changes in muscle activation, versus those who better maintained kinematics had minimal changes with altered shoes, and perhaps lesser changes with shod vs. barefoot. The study is generally well designed and written with findings presented in an appropriate manner.

General

Perhaps a matter of style, but in this reviewers’ opinion the manuscript is slightly askew in style. Specifically, the abstract is completely devoid of data, not even a mention of a % difference in muscle activity between those with “large” or small deviations in kinematics with alteration in shod status. Second, the introduction seems about a page too long, and the discussion a tad short.

Further, the abstract is quite vague and hard discern what the authors actually did, specifically line 24 “when a novel movement path was chosen”. I would strongly encourage the authors to better clarify here and elsewhere. To me, you stratified based upon kinematic differences between shod and unshod running, and differences with running in different shoes exploring the corresponding muscle activation. Another example is line 31 “when running conditions were different” and “when running conditions were similar”. Be specific, please. In this line, running conditions refers to two different concepts: shod vs unshod, and differences in shoe. Grammatically there is no concern, content wise the information is unnecessarily obscured.

There is no obvious rationale for the shoes chosen/compared. Even if practical in nature please provide the readers some rationale as to why these three shoes, the descriptions are somewhat helpful but would be bolstered by additional information (e.g. exact models). To those who don’t partake in the ‘mizuniverse’, some context would be helpful.

It seems all conditions might be “altered” running condition, as they never ran in their “native shoe”. Please comment.

Were these shoes new and/or “broken in” before the first participant? Please clarify.

I assume all conditions were run in a single visit? This is unclear, please clarify.

I fundamentally struggle with the term “preferred movement path”, it seems it IS their movement path, whether it is preferred or not might be a matter of debate. As a relatively equal number of men and women were recruited, an assumed differing Q angle of the women might be playing into differing kinematics and movement patterns which has been shown to relate to injury. Moreover, perhaps independent of sex, running kinematics likely plays a role in injury development (https://link.springer.com/article/10.1007/s42452-019-0695-x ), is their movement pattern preferred if it is predisposing them to an injury? I understand the sentiment, but preferred almost seems unnecessary, at least in this context or even misleading. Further in a number of instances the authors outrightly state athletes “choose” their PMP. Are the authors suggesting this is a conscious decision? Did you survey the athletes to ask them about their decisions? Kidding aside, their movement pattern might be also the result of a prior injury or surgery and not necessarily to avoid an injury (line 258), which could not be captured in the 6-month screening criteria. Individual movement patterns also likely depend upon sex/hormonal status, which is ignored in the current paper. The authors model proposes far more voluntary thought than is likely occurring, unless I under-think when I run.

The paradigm suggests optimization to lower energy cost it is unfortunate to have missed the opportunity to quantify this in any form to substantiate these claims.

Line 169-170 the authors should provide rationale for this cutoff, perhaps citing the previous work of one of the authors. Presently it appears arbitrary, which is confirmed in the discussion (line 300).

The statistical approach is somewhat confusing. If the authors are seeking to compare the groups, then shouldn’t this factor be included in a singular model rather signed rank tests “separate for each group”, such as a two-way ANOVA, to gain insight into potential interactions of group and running task? Based upon this the authors haven’t factually compared the groups and should refrain from such comparisons.

How is an absolute change a % when based upon change in EMG AUC? Include the calculation or perhaps the author mean “relative change”? Further, the data analysis section doesn’t seem to match the results section.

Minor

Line 253 “patter” to pattern

Line 310 “constrains” constraints

In the acknowledgements it seems there is a remnant from an MSSE submission, please remove.

Recent work in women suggests an increase in EMG with barefoot running as compared to shod (PMID 31839842), and seems a relevant reference to include.

Reviewer #2: General Comments:

This study was on whether muscle activity alterations can be a reflection of changes in movement path or running shoe conditions. This study was designed to test a new paradigm of running mechanics that explains that the cause of injury may not necessarily be directly associated with impact forces or increased pronation but could be the result of elevated changes in internal forces. Overall, the structure and style of this manuscript was very good. It was easy to follow and presented a new, interesting idea on running biomechanics. I had very few concerns about this paper, which are discussed below.

Specific Comments:

METHODS

Line 119: Was there rationale for only including heel-to-toe runners in this study? It was discussed later on that you wanted to have a study that consisted of a heterogeneous population, so I was wondering why forefoot or midfoot runners were not also included. I know the study was looking at EMG changes right before and after heel-strike, but I was thinking this paradigm could hopefully be tested and applied to all types of runners (i.e. striking patterns and running ability).

DISCUSSION

Line 244: You discussed that the preferred movement path paradigm has been proposed as a replacement of the traditional paradigms, but starting at line 54 in the introduction, you stated that this new paradigm has been introduced to further enhance traditional understanding. I was just confused by this connection because of the wording. Is this novel paradigm designed to replace or strengthen the previous paradigms?

Line 253: change ‘patter’ to ‘pattern’

Line 256: What was the purpose of using ‘most economical’ when describing the reasoning for why a preferred path is important? Does ‘economical’ refer to max VO2 or energy consumption? If it does not refer to just energy efficiencies, what does it entail? The reason I ask is because a few sentences following, you discussed that one of the reasons why a preferred movement path is selected is to reduce energy consumption, while in Line 256 you state that it is not directly correlated with running economy. I may be misinterpreting the statement, but I associate a higher running economy with lower levels of energy consumption.

Reviewer #3: General Comments

This manuscript attempts to shed light on the preferred movement path, a novel idea which may help with understanding relationships between running shoe design, running mechanics, and injury. Given the topic, this would be of interest to readers and important for the field. That said, I have some comments and concerns before this manuscript could be accepted for publication.

This is a revised version. While it is much improved over the previous versions, the authors have not utilized the previous reviewers comments fully to improve the manuscript. Rather, there are superficial acknowledgements of the reviewer comments and superficial fixes instead of meaningful discussion. More specifically, the authors have not adequately addressed the following:

• Reviewer 1’s question regarding how the use of the preferred movement path could inform shoe design and/or reduce incidence of running injuries.

• Both reviewer 1 and 2 requested more details regarding how outliers were handled and criteria for removing data. This has not been provided.

• Reviewer 1’s request for the authors to comment on whether similar results would occur following an adaptation period.

• Reviewer’s 2 suggestion to focus on the shod vs barefoot instead of the shod vs shod comparisons. The authors answer this saying this has already been done, but it has not since it is not possible to actually determine which movement path is the preferred.

• Reviewer 2’s comment about the EMG data analysis. The authors have not simplified the analysis or presented it in a way most readers would understand. They certainly did not do the analysis as reviewer 2 suggested. If they are not then at a minimum in the methods they need to justify why their approach is more insightful or a more appropriate approach.

• Reviewer 2’s comment about the results section being insufficient. The authors have superficially addresses this but still do not provide mean values or measures of effect size.

• Reviewer 2’s concern about potential bias given that the authors are the only group who has investigated or suggested the preferred movement path. This requires substantial attention and should be discussed openly in the limitations of this manuscript.

• Reviewer 2’s comment regarding more details being needed regarding the kinematic analysis. I am in agreement with reviewer 2 that it is not sufficient to simply cite the previous paper. This should stand on its own as an independent work. Additional details should be provided.

In addition to the above comments, I have several general comments that were not raised previously which I believe need to be addressed before this can be published.

• Related to reviewer 2’s concern about potential scientific bias, there is another group which is working on a hypothesis similar to the PMP. While the authors have cited Trudeau and colleagues first paper, they have done so in a very superficial manner and not incorporated the main points of Trudeau et al.’s work. This is an important element to discuss as, in contrast to the PMP, Trudeau and colleagues suggest a method for quantifying the habitual movement path and for assessing how much a shoe promotes or changes this path. They have also recently published a paper showing that a long run in shoes which do not promote the habitual movement path results in changes to cartilage volume as measured by MRI. At a minimum this new paper should be included in the introduction, most likely around the discussion in lines 62-64.

• The reviewers discuss the use of changes greater or less than 3 degrees as a cutoff for deciding whether individuals changed their movement path. Why have the authors not used some form of entire curve fitting comparison? It would seem that by using a somewhat arbitrary magnitude value the authors could end up with a participant in the substantially changes group where really the only thing that changed is the magnitude of the movement. Alternatively, the entire curve could still be shifted. However, the curve itself may have a very similar overall shape and trajectory. Indeed, this appears to be the case in most of the example curves presented in Figure 2, which should show the different movement paths. Really the ankle frontal plane motion is the only curve which looks different across conditions. All the other curves are simply offset.

• Did all runners maintain a consistent foot strike pattern in the barefoot and shod conditions? If not one could reasonably expect different kinematics as this has been shown many times. But is that really showing a difference in preferred pattern, or simply a difference between footwear conditions?

• How do the participants in Table 1 match with the data presented in Figures 4 and 5? It is not clear which participants from which groups are shown in Figures 4 and 5.

Specific Comments:

Abstract, lines 30 – 37. These are wordy and difficult to follow as currently written. Suggest revising for clarity.

Line 42 – What do the authors consider “excessive” impact forces or pronation. While their point is well made that these are the dominant paradigms, is the problem really with the paradigms or with the use of the term “excessive” amount where there is no agreement on what constitutes excessive? Perhaps higher levels would be a better phrase?

Line 59 – Here the authors suggest runners maintain a given movement path when changing between different shoe conditions. Yet, in line 67 the authors suggest that the movement pattern might be different in running shoes verse ski boots. These appear to be presenting opposite views.

Line 111 – What kinematic differences are the authors referring to? The entire curve? Peak values, range of motion? Some more detail is needed.

Lines 122 – 123 – The authors have opened the study to a wide cross section of runners. While this improves the external validity, the authors should also report the experience level of their participants. How much mileage were they running each week? What was average training paces? This would be especially important information to present in light of the discussion in lines 291 – 299 about experience level.

Acknowledgements – There appears to be verbiage in here from a previous submission to an ACSM journal. If this is not required for PLoS One then it should be removed.

6. PLOS authors have the option to publish the peer review history of their article (what does this mean?). If published, this will include your full peer review and any attached files.

Reviewer #1: No

Reviewer #2: No

Reviewer #3: No

---

## [Author Response · Author response to Decision Letter 0]

28 May 2020

All comments have been replied to in the separate document titled response to the reviewers. We copied the original comment of the reviewers and answered in red text below it for your convenience.

Further, Figure 3 is now referred to within the text (Line 185) and a statement to reflect this change has been added in the 'response to the reviewers' document.

---

## [Decision Letter · Decision Letter 1]

14 Aug 2020

PONE-D-20-04762R1

Influence of running shoes on muscle activity

PLOS ONE

Dear Dr. Hoitz,

Thank you for submitting your manuscript to PLOS ONE. After careful consideration, we feel that it has merit but does not fully meet PLOS ONE’s publication criteria as it currently stands. Therefore, we invite you to submit a revised version of the manuscript that addresses the points raised during the review process. One of the reviewers requested that you make a very minor edit (see details below).  

We look forward to receiving your revised manuscript.

Kind regards,

Jean L. McCrory, PhD

Academic Editor

PLOS ONE

Reviewers' comments:

Reviewer's Responses to Questions

**Comments to the Author**

1. If the authors have adequately addressed your comments raised in a previous round of review and you feel that this manuscript is now acceptable for publication, you may indicate that here to bypass the “Comments to the Author” section, enter your conflict of interest statement in the “Confidential to Editor” section, and submit your "Accept" recommendation.

Reviewer #1: (No Response)

2. Is the manuscript technically sound, and do the data support the conclusions?

Reviewer #1: Yes

3. Has the statistical analysis been performed appropriately and rigorously? 

Reviewer #1: (No Response)

4. Have the authors made all data underlying the findings in their manuscript fully available?

Reviewer #1: (No Response)

5. Is the manuscript presented in an intelligible fashion and written in standard English?

Reviewer #1: (No Response)

6. Review Comments to the Author

Reviewer #1: The authors have made significant improvements to the manuscript. However, a minor issue remains. Namely, in line 367 " ...in [ previous work?]. This needs an edit.

7. PLOS authors have the option to publish the peer review history of their article (what does this mean?). If published, this will include your full peer review and any attached files.

Reviewer #1: No

---

## [Author Response · Author response to Decision Letter 1]

19 Aug 2020

Thank you for your assessment. We greatly appreciate that our efforts to improve this manuscript have been well received. Following the reviewer’s suggestion, we have updated the respective line. The clean version of the manuscript now reads: ‘This threshold was selected as it represents a conservative threshold for clinical relevance and was suggested in previous work [10]’ (Line 291).

For your convenience, we highlighted this change in the version with tracked changes.

---

## [Editor Report · Decision Letter 2]

15 Sep 2020

Influence of running shoes on muscle activity

PONE-D-20-04762R2

Dear Dr. Hoitz,

We’re pleased to inform you that your manuscript has been judged scientifically suitable for publication and will be formally accepted for publication once it meets all outstanding technical requirements.

Kind regards,

Jean L. McCrory, PhD

Academic Editor

PLOS ONE

---

## [Editor Report · Acceptance letter]

28 Sep 2020

PONE-D-20-04762R2 

Influence of running shoes on muscle activity 

Dear Dr. Hoitz:

I'm pleased to inform you that your manuscript has been deemed suitable for publication in PLOS ONE. Congratulations! Your manuscript is now with our production department. 

Kind regards, 

on behalf of

Dr. Jean L. McCrory 

Academic Editor

PLOS ONE